# Effect of High-Intensity Interval Training Combined with Fasting in the Treatment of Overweight and Obese Adults: A Systematic Review and Meta-Analysis

**DOI:** 10.3390/ijerph19084638

**Published:** 2022-04-12

**Authors:** Zhicheng Guo, Jianguang Cai, Ziqiang Wu, Weiqi Gong

**Affiliations:** School of Physical Education, Hunan University of Science and Technology, Xiangtan 411201, China; 20021701003@mail.hnust.edu.cn (Z.W.); 20011701005@mail.hnust.edu.cn (W.G.)

**Keywords:** high-intensity interval training, fasting, obesity, treatment

## Abstract

Objectives: A systematic review and meta-analysis is conducted to compare the effects of high-intensity interval training (HIIT) combined with fasting (HIIT + fasting) and other interventions (HIIT alone, fasting alone, or normal intervention) in adults with overweight and obesity on body composition (body mass, body mass index (BMI), waist circumference (WC), percent fat mass (PFM), fat mass (FM), fat-free mass (FFM)), maximal oxygen uptake (VO_2peak_), and glucose metabolism (fasting plasma glucose (FPG)), fasting plasma insulin (FPI)). Methods: The databases of PubMed, the Cochrane Library, Embace, Web of Science, CNKI, Wangfang Data, and CBM were searched from their inception to February 2022. Randomized controlled trials comparing the effects of HIIT + fasting and other interventions on adults with overweight and obesity were included in this meta-analysis. The risk of bias was assessed by the Cochrane risk of bias tool. The effect size was completed by using mean difference (MD) and standard deviation. If there were varying units or large differences among the included studies, the standardized mean difference (SMD) would be used. The certainty of evidence was evaluated using the Grading of Recommendations Assessment, Development, and Evaluation (GRADE). Results: Nine randomized controlled trials with 230 overweight and obese adults were conducted in accordance with our inclusion criteria. The results of the meta-analysis revealed that compared to the control group HIIT + fasting had better effects on the body mass, WC, FM, and VO_2peak_, while there were no significant differences in PFM, FFM, FPG, and FPI. Conclusions: Despite the number of included trials being small and the GRADE of all outcomes being very low, HIIT + fasting has a positive effect on the body composition of overweight and obese adults, and significantly improves VO_2peak_. For adults with overweight and obesity who have long-term comorbidity, HIIT + fasting was a better way to improve FPG than HIIT alone or fasting alone. More studies are required to investigate different combinations of HIIT + fasting; and the safety of HIIT + fasting intervention on overweight and obese adults.

## 1. Introduction

Obesity has evolved into a global public health issue, and the proportion of the obesity in population is increasing. Many guidelines define obesity as a body mass index (BMI) above 30 kg/m^2^ [1]. Compared with normal people, people with obesity have a higher risk of developing chronic diseases [2,3,4]. Several studies have regarded obesity as a chronic disease that needs to define clinical strategies and structured payment policies to combat [5,6].

Exercise is currently recognized as one of the best measures to treat obesity because it can improve body composition and enhance aerobics capacity [7]. Much research has documented that moderate physical activity has a positive effect on losing weight, reducing central adiposity, and preventing obesity [8,9]. A recent systematic review reveals that training brings more reduction on the total body mass (especially visceral adiposity) compared to a hypocaloric diet [10]. Berge et al. [11] show that different intensity aerobic activity has similar effects on severe obesity in increasing energy expenditure and inducing moderate weight loss. Zhang et al. [12] also found that 6 months of vigorous training (150 min 65–80% of maximum heart rate (HR_max_)) and moderate training (150 min 45–55% of HR_max_) are equal in reducing intrahepatic triglyceride content.

In recent years, numerous clinical experiments have demonstrated that fasting is an effective way to reduce fat in obese people [13]. Intermittent fasting has a positive impact on the treatment of chronic diseases, including diabetes, cardiovascular disease (CVD), cancer, etc. [14]. Energy-restricted diets of different meal frequencies have equal effects on body weight, total fat mass, and hepatic fat reduction [15]. In addition, Ramadan intermittent fasting (RIF) can both improve systemic inflammation biomarkers [16] and appetite-regulating hormones [17] in males with obesity.

The comparison of exercise and fasting effects on obesity has been studied by many scholars. A recent finding has suggested that exercise is better than a calorie-restriction program in cholesterol biosynthesis, no matter with or without alternate-day fasting (ADF) [18]. Short-term exercise in conjunction with diet interventions has great impacts on decreasing metabolic risks and fasting insulin levels in children with obesity [19].

High-intensity interval training (HIIT), as a time-saving and effective workout method, has been extensively promoted for mass fitness and disease prevention. Compared to traditional exercise, HIIT appears to be risk-free and has higher compliance [20]. A recent meta-analysis suggests that HIIT is more effective and time-efficient than other types of exercise on blood pressure and aerobic capacity [21]. Wu et al. [22] showed that HIIT can also delay muscle atrophy and enhance aerobic exercise capacity in the old. Recent meta-analyses also reveal that HIIT is a high-efficiency program for the reduction of abdominal and visceral fat mass in adults [23]. There is a consensus that HIIT can elicit similar or better effects on obesity than traditional physical activities (i.e., moderate-intensity continuous training (MICT)) [24]. However, whether the combination of HIIT and other interventions could bring more meaningful improvement requires more evidence to prove.

Fasting is proven to have an advantage in resisting metabolic syndrome, chronic age-related disorders, and obesity-related disorders [25,26]. In the short term, fasting is an efficient tool for losing weight, which can achieve a substantial weight loss for a few weeks [27]. In terms of CVD, fasting can significantly reduce the risk factors with or without weight loss [28]. Different types of fasting programs can bring various benefits. A recent finding from a meta-analysis demonstrates both ADF and continuous energy restriction can reduce the body mass, BMI, fat mass (FM), and total cholesterol of overweight adults, and ADF does better in waist circumference (WC) in ≥40 years adults [29]. Although fasting is often combined with exercise, the combination of HIIT and fasting is a new tool for weight loss, which needs more attention.

There is no consensus on the therapeutic effect of HIIT combined fasting (HIIT + fasting) in adults with obesity, and no consensus as to whether the weight reduction of HIIT + fasting is superior to HIIT alone or fasting alone. Therefore, we aim to compare the impact of HIIT + fasting and other interventions (HIIT alone, fasting alone, or normal intervention) on body composition in the obese and overweight population by performing a meta-analysis. Secondary aims were to determine which combination method has a better effect on adults with overweight and obesity.

## 2. Method

The meta-analysis protocol has been registered with the International Prospective Register of Systematic Reviews (registration number: CRD42021283287), and the meta-analysis was conducted in accordance with the PRISMA-P guidelines [30].

### 2.1. Inclusion and Exclusion Criteria 

As Table 1 shows, there was no limitation on comparison. The following criteria were used to screen studies: (1)Participants: The participants were ≥18-year-old adults diagnosed with overweight or obesity. The definitions of overweight and obesity were based on age and sex-specific BMI cut-off points as indicated by WHO (overweight, ≥25 kg/m^2^; obesity, ≥30 kg/m^2^); age-incompatible, normal-weight adults, and animal-based subjects were excluded (considering that obese patients often have complications and some obesity is disease-caused, the participants with complications will also be included and illustrated in the results).(2)Intervention: Only participants who received HIIT + fasting were included in this study. HIIT intervention measures were used: the exercise intensity at 80 to 100% HR_max_/peak oxygen consumption (VO_2peak_) for 30 s to 4 min, passive recovery, or low-intensity aerobic exercise in the intermittent period for a maximum of 4 min [31]. The frequency of HIIT was more than twice a week. Fasting programs involved different forms of fasting and energy-restricting diets (e.g., low-calorie diet (LCD), low-carbohydrate diet, intermittent fasting, and intermittent energy restriction) [32].(3)Study: Randomized controlled trials (RCT) were included.(4)Outcomes: Data types related to body composition, glucose metabolism, and cardiorespiratory fitness were included. Body composition: Body mass, BMI, WC were objectively measured on a digital scale, a stadiometer and a plastic tape (WC was measured 2 cm above the umbilicus). FM and fat-free mass (FFM) were measured on hydrostatic weighing, dual-energy X-ray absorptiometry (DXA), bioelectrical impedance analysis (BIA), computed tomography (CT), or magnetic resonance imaging (MRI) [33]. Percent fat mass (PFM) was measured indirectly. Cardiorespiratory fitness: VO_2peak_ was assessed by a stepwise cardiopulmonary exercise test (CPET) by bicycle, a continuous incremental test on a cycle ergometer with indirect calorimetry, or the Vmax Encore System (CareFusion Corp., San Diego, CA, USA). Glucose metabolism: Fasting plasma glucose (FPG) was measured by a kit assay (Pointe Scientific, Canton, MI). Fasting glucose insulin (FPI) was analyzed by ELISA (ALPCO Immunoassays, Salem, NH, USA).

### 2.2. Literature Retrieval Strategy

Following the guideline of the PRISMA statement, we carried out a thorough search of the electronic literature. Relevant studies published and unpublished which tested the association between HIIT, fasting, and fat reduction were searched, with a language restriction of English and Chinese. Seven electronic databases (Pubmed, Embase, Cochrane library, Web of Science, CNKI, Wanfang, and CBM) were searched from inception to 24 February 2022. We also looked through the list of eligible research cited in the relevant journals and references to see if any more studies were potentially suitable. The details of the search criteria used are described in Table 2.

### 2.3. Literature Selection

Two researchers (Guo and Wu) independently performed a literature-screening process following the PRISMA guidelines. The duplicate entries were removed by using a bibliographic reference manager (EndNote X9). Research that did not fit our inclusion requirements was ruled out by reading the titles and abstracts. The remaining studies were evaluated by reading the full texts. Any anomalies in the screening process were worked out with the help of a third researcher through conversation or consultation.

### 2.4. Data Extraction

The key data extraction was carried out independently by two reviewers. The following data of each included study were extracted: first author’s name, year of publication, country, participant population, participant characteristics, details of intervention (e.g., HIIT intensity and time, fasting strategy). For main continuous outcomes (the number of participants, mean, and standard deviation (standard error or 95% confidence interval)), we extracted from the follow-up immediately after the end of the intervention. Any disagreements were resolved by consensus. The data extraction was used a modified version of the Cochrane Collaboration data collection form for intervention reviews: RCTs only [34]. We emailed the corresponding authors of the studies for data that could not be extracted from the published studies. If the authors did not reply within 15 days of our emails, we considered the data as missing.

### 2.5. Risk of Bias Assessment

The critical appraisal was carried out by two researchers (Guo and Wu) and the risk of bias assessment of the included RCTs was using a “risk of bias” approach, which is recommended by the Cochrane risk of bias tool [35]. Studies that met the inclusion criteria were checked for random sequence generation, allocation concealment, blinding, incomplete outcome data, selective reporting, and other biases; each evaluation result was rated as “high risk, low risk, and unclear based” on related criteria.

### 2.6. Statistical Analysis

In this study, Revman 5.3 and Stata 15 were used to carry out a meta-analysis. The effect size between pre-intervention and post-intervention in each group was completed by using the mean difference (MD) and standard deviation. If there are varying units or large differences among the included studies, the standardized mean difference (SMD) with 95% CI was used. If the baseline of the HIIT + fasting group and control group were close (*p* > 0.05), we would use the effect size of the post-intervention. The changes between baselines and outcome would be used, while the differences from the baseline were significant (*p* < 0.05). Considering that the included RCTs and the fasting plans would have great differences, the random effect model was adopted for all the outcomes. If heterogeneity is moderate or high (50% ≤ I^2^ < 75% or 75% ≤ I^2^ < 100%), the sensitivity analysis would be carried out. Moreover, subgroup analysis was performed to find out whether different binding forms of HIIT + fasting can elicit a beneficial effect on fat reduction. Duration of intervention, interval protocol, and form of fasting were examined as subgroups. The subgroup analysis of each outcome must include at least two studies, and the heterogeneity between subgroups was assessed by a chi-square test. Egger’s test was used for publication bias analysis. Because the number of studies is too small (<10), univariate meta-regression analyses were not performed.

### 2.7. Certainty Assessment

The certainty of evidence was assessed using the Grading of Recommendations Assessment, Development, and Evaluation (GRADE). A ‘summary of findings’ table was created according to the GRADE approach to make clinical practice recommendations [36]. For all outcomes, two researchers (Wu and Gong) independently completed the assessment. Any disagreements were solved by a third researcher (Cai) through conversation. The certainty of the evidence was graded as high, moderate, low, or very low. RCTs received a high grade as the default and were downgraded according to the following criteria: risk of bias (the result of the Cochrane risk of bias tool), indirectness (unexplained heterogeneity between including trails, I^2^ > 50% and *p* < 0.01), imprecision (the 95% CI of effect size were wide, number of participants < 400)), and publication bias (significant evidence of small-study effects).

## 3. Result

### 3.1. Study Identification and Selection

In total, 521 related RCTs were identified after searching the database. Then the duplicates were removed by using EndNote X9. The titles and abstracts of the remaining 236 studies were evaluated by two reviewers independently, and 197 were removed. The full texts of the last 39 studies were reviewed diligently. Seventeen studies were excluded because they did not meet the inclusion criteria, six for poor quality, and seven for missing data. Nine RCTs, [37,38,39,40,41,42,43,44,45], were finally considered eligible in accordance with our inclusion criteria. (More details were shown in Figure 1).

### 3.2. Characteristics of Included Studies and Participants

Our evaluation criteria were met by 230 overweight and obese participants (age ≥ 18). The primary characteristics of nine RCTs were shown in Table 3. In total, 124 participants were allocated in the HIIT + fasting group, and the remaining were allocated in the control group (HIIT alone, fasting alone, or normal intervention).

All of the participants underwent HIIT by bicycle [37,38,39,40,41,42,43,44,45]. Five studies used an LCD as a fasting program [37,38,39,40,41]. Three used a low-carbohydrate diet [40,44,45]. Only one study used other fasting plans [39]. Training intensity was assessed using HR_max_ or VO_2peak_. The nine included studies could be divided into short-term HIIT + fasting and long-term HIIT + fasting. The duration of the short-term HIIT + fasting was within 13 days [37,38,41,42,44]. The duration of the long-term HIIT + fasting ranged from 1 month to 1 year [39,40,43,45]. The majority of the control programs underwent fasting alone [37,38,40,41,42,44]. Two control programs only underwent HIIT [39,43]. Only Sun [45] did not report the intervention programs of the control groups. Most HIIT was performed three times a week. Regarding complications, the participants of Gyorkos [40] had metabolic syndrome and the participants of Pedersen [43] had CAD.

### 3.3. The Risk of Bias Assessment

As Figure 2 and Figure 3 presented, nine trials were evaluated for risk bias. Only two RCTs reported the means of allocation concealment. Since it was nearly impossible to keep the participants and staff in the dark about the intervention, all trials were evaluated as “high risk of bias”. The outcome assessment of all studies was unknown. There were no significant biases on incomplete outcome data, selective reports bias, or other biases among the included trials. The main bias was due to the differences in fasting programs and the intervention of the control group among the included studies. The duration of HIIT + fasting and the interval of HIIT also led to the bias in articles.

### 3.4. Meta-Analysis

All studies reported the effect of HIIT + fasting on body composition. Nine RCTs reported post-intervention in body mass, seven RCTs reported BMI, six RCTs reported WC, six RCTs reported PFM, six RCTs reported in FM, and seven RCTs reported in FFM. There were no significant differences (*p* > 0.05) found in the baseline of the body mass, BMI, WC, PFM, FM, and FFM, so we used the effect size of the post-intervention. A random effect model was used with the effects measures of MD for body mass, WC, PFM, FM, and FFM, and SMD for VO_2peak_, FPG, and FPI.

From the meta-analysis, significant differences were found for body mass (MD = −2.97 kg, 95% CI: −5.83 to −0.12, *p* = 0.04), BMI (MD = −1.22 kg/m^2^, 95% CI: −2.31 to −0.13, *p* = 0.03), WC (MD = −4.33 cm, 95% CI: −7.11 to −1.55, *p* = 0.002), and FM (MD = −2.18 kg, 95% CI: −3.84 to −0.51, *p* = 0.01) in HIIT + fasting group relative to control group (fasting alone or HIIT alone). No significant differences were observed for PFM (MD = −0.90%, 95% CI: −2.20 to 0.39, *p* = 0.17), and FFM (MD = 0.32 kg, 95% CI: −2.01 to 2.66, *p* = 0.79), which might have been because HIIT + fasting reduced the total body composition (Figure 4).

In terms of cardiorespiratory fitness, the changes of HIIT + fasting on VO_2peak_ were reported in seven RCTs. Compared to the control group (fasting alone or HIIT alone), there was a meaningful improvement in the HIIT + fasting group on VO_2peak_ (SMD = 0.78, 95% CI: 0.42 to 1.14, *p* < 0.00001). The finding meant that HIIT + fasting was much better than HIIT alone or fasting alone intervention in cardiorespiratory fitness (Figure 4).

Regarding glucose metabolism, seven RCTs measured the changes in FPG. Six RCTs measured the changes in FPI. The combined findings of the RCTs revealed that HIIT + fasting had no significant impacts on FPG levels (SMD = −0.12, 95% CI: −0.51 to 0.27, *p* = 0.54) or FPI (SMD = 0.13, 95% CI: −0.21 to 0.47, *p* = 0.86) relative to the control group (fasting alone or HIIT alone) (Figure 4).

For sensitivity analysis, we eliminated RCT respectively to ensure the reliability of the research results. No significant changes were found in the effect size, which indicated that the findings of the meta-analysis were reliable.

### 3.5. Subgroup Analysis

According to the HIIT + fasting duration, interval protocol, and form of fasting, we divided the HIIT + fasting program into “long term: the duration of HIIT + fasting ≥1 month, and short term: <1 month; long interval: intermittent time 1–3 min, and short interval: 10–60 s. LCD: use LCD as fasting plan, and low-carbohydrate diet: use a low-carbohydrate diet as fasting plan for subgroup analysis”. From the analysis, HIIT + fasting of ≥1 month had positive effects on most outcomes in body composition compared with the control group (fasting alone or HIIT alone), except the PFM (MD = −1.04%, 95% CI: −2.84 to 0.76, *p* = 0.26) and FFM (MD = −2.28 kg, 95% CI: −6.38 to 1.82, *p* = 0.28). Both HIIT + fasting of ≥1 month and of <1 month had significant benefits on VO_2peak_. HIIT + fasting of ≥1 month also had effects on FPG (SMD = −0.61, 95% CI: −1.09 to −0.13, *p* = 0.01), which HIIT + fasting of <1 month did not. Moreover, neither long-interval or short-interval had any significant improvement on PFM (long-interval, MD = −0.61%, 95% CI: −2.41 to 1.19, *p* = 0.51; short-interval, MD = −1.21%, 95% CI: −3.07 to 0.64, *p* = 0.20). Different from long-interval, short-interval HIIT + fasting showed no positive benefits on VO_2peak._ However, short-interval HIIT + fasting had positive effects on FPG (SMD = −0.61, 95% CI: −1.09 to −0.13, *p* = 0.01). HIIT + fasting of a low-carbohydrate diet also had positive effects on FPG (SMD = −0.58, 95% CI: −1.06 to −0.09, *p* = 0.02), which was greater than the fasting or HIIT alone (Table 4).

### 3.6. GRADE Assessment

The certainty of evidence for the outcomes body mass, BMI, WC, PFM, FM, FFM, VO_2peak_, FPG, and FPI was assessed using the GRADE system and presented through the Summary of Findings Table 5. For all outcomes, the quality of the evidence was very low. Most downgrades were due to the risk of bias, limitation of participants, and different fasting plans. The downgrades of PFM, FFM, FPG, and FPI were due to the confidence interval reaching the null effect.

## 4. Discussion

Obesity and overweight have become major risk factors threatening human health in the 21st century [46] and pathogenic factors for various chronic diseases. There is a dose-effect link between the quantity of exercise and health status [47]. Many studies have confirmed that HIIT with different exercise intensity, time, and frequency combinations has advantages in fat and weight reduction [6,48]. The importance of HIIT combined with dietary intervention has been known, and some researchers proved that the efficacy of HIIT on fat and weight reduction is not significant in the absence of dietary control [49,50]. To our knowledge, this review is the first research to directly analyze the fat and weight reduction effect of HIIT + fasting, even though there were no significant differences found for PFM or glucose metabolism between HIIT + fasting and control groups (fasting alone or HIIT alone). The results showed that HIIT + fasting was an efficient intervention in improving the body composition. Furthermore, the changes in VO_2peak_ following HIIT + fasting were substantially larger than the changes in HIIT alone or fasting alone at the same session. Considering the intervention included the fasting plan, we also compared the differences between HIIT + fasting and the control group (fasting alone or HIIT alone) on FPG and FPI, and there were no significant differences found. However, we found that HIIT + fasting of ≥1 month, short-interval, or a low-carbohydrate diet has more positive effects on FPG. These findings indicated that compared to traditional intervention HIIT + fasting is more effective in reducing body fat and increasing aerobic capacity.

The results demonstrated that HIIT + fasting can elicit a meaningful impact on body mass and BMI, compared to the fasting group or the HIIT group. Bodyweight and BMI were reduced by −2.97 kg and −1.22 kg/m^2^, respectively. These findings were in accordance with those reported by Pedersen et al. [43], whose trials explored the effect of HIIT + fasting on overweight adults with CAD and demonstrated that compared to the HIIT group HIIT + fasting has a long-term benefit in body composition.

The subgroup analysis showed that in body composition there were no greater changes in short-term HIIT + fasting than in long-term HIIT + fasting, which is similar to HIIT intervention [51]. The changes in body mass and BMI in short-term HIIT + fasting were 0.52 kg and −0.15 kg/m^2^, which meant that there were no differences between the HIIT + fasting group and the control group (fasting alone or HIIT alone) on the benefits of body mass and BMI. The result reported by Taylor et al. [52] whose trials found that the effects of ≥3 months HIIT on body mass and BMI were much better than 4 weeks HIIT, the similar results were also found in fasting intervention [53]. These findings suggested that it took over 1 month for HIIT + fasting to improve body mass and BMI, which could become a guideline of HIIT + fasting in clinical application. Our finding showed that compared to the control group, HIIT + fasting had no significant effects on PFM, which is similar to Viana et al. [54] whose meta-analysis that combined 24 studies revealed that different interventions had almost no differences in the improvement of PFM. The combination of the different interventions seemed to have little clinical significance on PFM reduction. Previous research results have demonstrated that either HIIT [55,56] or fasting [57] has beneficial impacts in reducing body mass and BMI, and our results suggested that HIIT + fasting might achieve a better effect than traditional HIIT or fasting intervention.

Abdominal obesity is a manifestation of visceral obesity, and WC is considered to be a criterion for evaluating abdominal obesity [58,59]. The findings of our analysis showed that HIIT + fasting was more effective in improving the WC of adults with overweight and obesity than the control group (HIIT alone or fasting alone). The subgroup analysis showed that except for the short term almost all the subgroups have >4 cm WC more reduction. Although in the long-term, long-interval, and LCD subgroups the difference between the HIIT + fasting group and the control group were all found statistically significant. We still suggested HIIT + fasting should last for at least 1 month to achieve meaningful WC reduction because the included studies in the long-interval and LCD subgroups are the same. The reduction of the long-term group was more than the long-interval group in WC, and this finding demonstrated that in HIIT + fasting intervention the duration might be more meaningful than the interval. Similar results were also found in HIIT [52] and fasting [60]. Although compared to the fasting or HIIT intervention, the reduction in the short-term group and short interval group were not statistically significant, the WC of overweight and obese adults still had a meaningful reduction. From these factors, we suggested that the clinic would achieve more reduction in WC no matter what type of HIIT + fasting was used.

Relative to BMI, FM and FFM have better diagnostic values on CVD and obesity [61,62]. Our study found that the FM of participants undergoing HIIT + fasting was improved more significantly than participants undergoing HIIT or fasting. However, such improvement was not found in FFM. Similar to a recent meta-analysis, which suggested that the combination of HIIT and low-carbohydrate high-fat diet (LCHF) has no significant effects on FM and FFM [63], we found there were no significant differences between HIIT + fasting and fasting in the low-carbohydrate subgroup. Given the differences in the HIIT protocol, we could only speculate that the combination of HIIT and LCD was a better intervention on FM than HIIT combined with a low-carbohydrate diet. Yancy et al. [64] showed that the LCD diet program has more meaningful improvement in body mass compared to the low-fat diet, and there were more improvements on serum triglyceride levels and high-density lipoprotein cholesterol in the LCD group than in the low-fat group. Therefore, we inferred that the meaningful reduction in HIIT + fasting on FM benefitted from the fasting program combined. Previous studies [65,66] have shown that fasting results in a loss of FFM in addition to weight reduction. From the subgroup analysis, we found that compared to the control group (fasting alone or HIIT alone) the FFM of short-term HIIT + fasting was better, while in long-term HIIT + fasting the result is exactly the opposite. These results indicated that HIIT was a way to improve the loss of FFM caused by fasting, and duration was a moderator for it.

Notably, the changes of VO_2peak_ in the HIIT + fasting group were better than the control group (fasting alone or HIIT alone), and the result of the meta-analysis was statistically significant (*p* < 0.0001). We were still not sure whether different interval or fasting programs could influence the positive effects of HIIT + fasting on VO_2peak_ because of high heterogeneity (I^2^ > 75%) in the short interval subgroup and low-carbohydrate diet. Similar findings were also reported by Wen et al. [67] whose analysis showed that different protocols of HIIT all have more beneficial effects than the control group on VO_2peak_. The finding of an RCT also revealed that the improvement of the HIIT program (exercise 30 s, interval 2 min) is similar to the MICT program (exercise 45–60 min) on VO_2peak_ [68]. To determine whether LCD had benefits on VO_2peak_, we checked all included studies whose control groups were LCD [37,38,41,42,43] and none of the results reported that there were significant changes in the LCD group. The meaningful improvement in cardiorespiratory fitness might be gained only through exercising.

Because fasting may have an impact on the digestive system, we carried out a meta-analysis to determine whether HIIT + fasting could cause more changes in glucose metabolism compared to the control group (fasting alone or HIIT alone). Although there were no significant changes found in FPG or FPI, we found that the differences in the long-term, short-interval, and low-carbohydrate diet subgroup were statistically significant. After further reading of the included studies, we found that HIIT + fasting had a more meaningful improvement than fasting on FPG and FPI in obese people who have a comorbidity [40,45]. Ryan et al. [69] demonstrated that only the most recent exercise sessions rather than long sessions of HIIT or MICT have insulin-sensitizing effects. Recent Studies [70,71] also suggested that the reduction of FPG in people with type 2 diabetes or the older in the HIIT group is caused by a low rate of exogenous glucose appearance. Cho et al. [72] revealed that the metabolic benefit of intermittent fasting may be caused by improving glycemic control, which HIIT was proved to have a similar effect [73]. From these findings, the effects of HIIT + fasting might profit from both HIIT and fasting and achieve more positive effects on FPG in people with overweight and obesity who have comorbidity.

To comprehensively analyze the effect of HIIT + fasting intervention, we investigated the duration of HIIT + fasting, the length of the interval, and the fasting plan HIIT combined. However, there were several flaws in this meta-analysis that must be addressed. First, although we used a broad search strategy to ensure all related articles were searched, only nine RCTs were considered to be eligible. Because of the small number of RCTs, Egger’s test and univariate meta-regression analyses were not carried out. Further analysis in the subgroups was also limited. Second, most of the included participants in our study were female, and gender differences might affect the accuracy of results. Third, most included HIIT + fasting programs combined HIIT with LCD. Only three studies used a low-carbohydrate diet, and only one study used time-restricted feeding. That might be because the LCD had better security and operability. Fourth, only two control groups of included RCTs were HIIT plans; whether HIIT + fasting had more improvement on overweight and obese adults compared to HIIT was hard to say. All the flaws lead to the main bias and the possible bias effects on the main evidence. The very low certainty of the evidence was also due to these reasons.

Despite these limitations, our study conducted a thorough analysis of the RCTs included, to investigate whether HIIT + fasting has more meaningful effects on overweight and obese adults than other interventions. The more time-saving and safer characteristics make HIIT + fasting a better treatment for the clinical treatment of obesity. In addition, the improvement of different fasting programs combined with HIIT on overweight and obese adults needed to be further explored. More studies with large samples and long duration are required to explore the effects HIIT + fasting can elicit on VO_2peak_ and glucose metabolism, improve the HIIT + fasting protocol, and find a better combination of HIIT + fasting to treat adults with overweight and obesity.

## 5. Conclusions

HIIT combined with fasting can effectively reduce body mass, BMI, WC, and FM of adults with overweight and obesity, and improve their VO_2peak_. HIIT + fasting of ≥1 month has a significant effect on weight loss. For adults with overweight and obesity who have comorbidity long term, HIIT + fasting was a better way to improve FPG than HIIT alone or fasting alone. These findings indicate that HIIT + fasting can be a better weight loss program than HIIT alone or fasting alone. However, more studies are still required to explore different combinations of HIIT and fasting, and the safety of HIIT + fasting intervention on adults with overweight and obesity.

## Figures and Tables

**Figure 1 ijerph-19-04638-f001:**
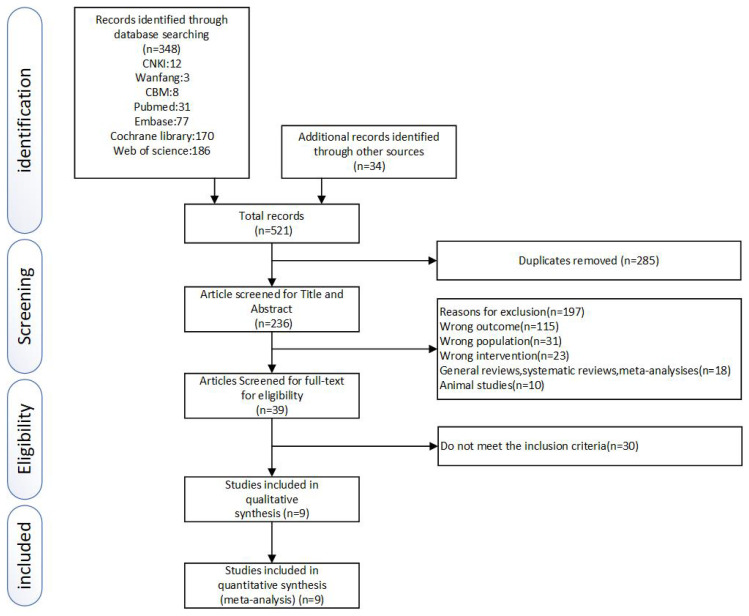
Literature search and study selection process.

**Figure 2 ijerph-19-04638-f002:**
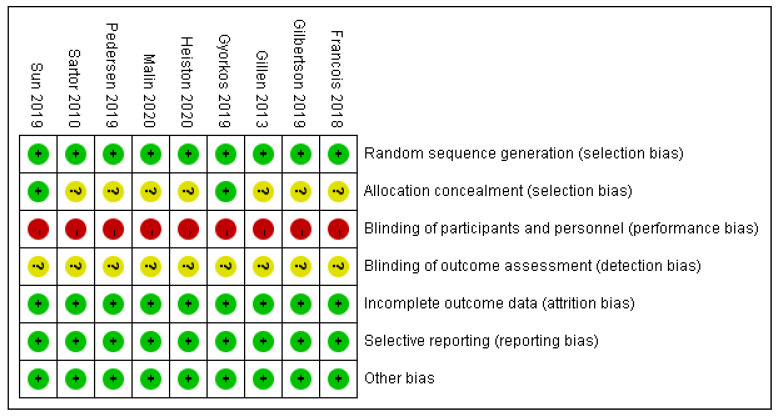
Review judgment of risk bias for each item: percentages across all included studies. Risk of bias levels: low (green or “+”), unclear (yellow or “?”), and high (red or “−”).

**Figure 3 ijerph-19-04638-f003:**
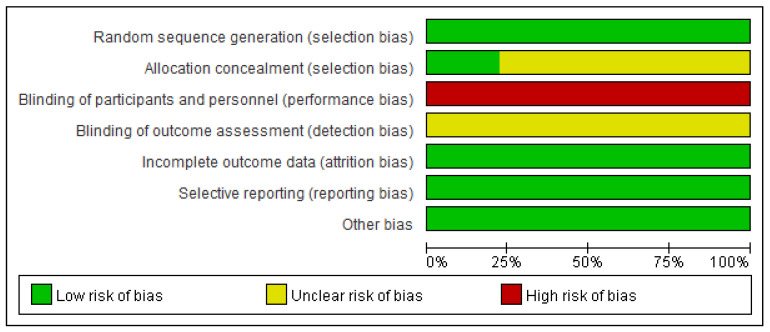
Summary of risk bias: review authors’ judgment of risk bias for each item. Risk of bias levels: low (green), unclear (yellow), and high (red).

**Figure 4 ijerph-19-04638-f004:**
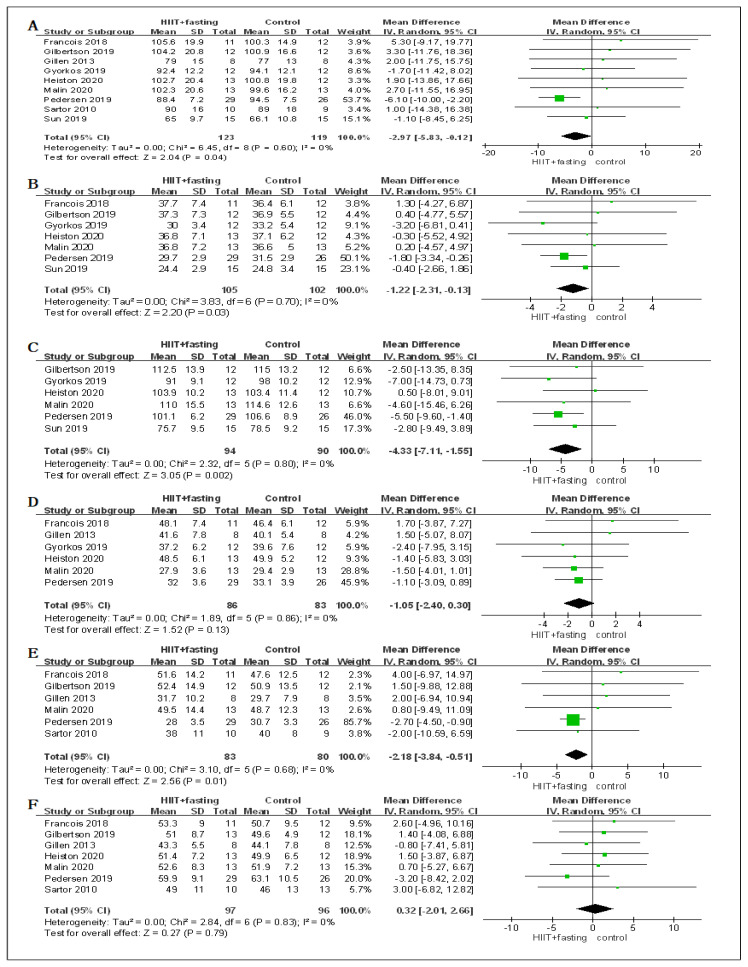
Forest plot for the body composition. (**A**) Forest plot for the body mass (BM); (**B**) Forest plot for the body mass index (BMI); (**C**) Forest plot for the waist circumference (WC); (**D**) Forest plot for the percent fat mass (PFM); (**E**) Forest plot for the fat mass (FM); (**F**) Forest plot for the fat-free mass (FFM); (**G**) Forest plot for changes in VO_2peak_.; (**H**) Forest plot for changes in fasting plasma glucose (FPG); (**I**) Forest plot for changes in fasting plasma insulin (FPI). The size of each small green box represents the relative weight of the studies conducted in the meta-analysis. The black diamond represents the results of the meta-analysis combining the individual studies.

**Table 1 ijerph-19-04638-t001:** Criteria for inclusion and exclusion.

PICOS	Inclusion	Exclusion
Participant	≥18 years old diagnosed with overweight or obesity	Age-incompatible; normal-weight; animal-based subjects
Intervention	HIIT + fasting	Other intervention
Outcome	BM; BMI; WC; PFM; FM; FFM; VO_2peak_; FPG; FPI	Other outcomes
Study	RCT	Books; opinion articles; observational studies; reviews; prospective cohort studies; studies and abstracts without adequate data

BM, body mass; BMI, body mass index; FM, fat mass; PFM, percent fat mass; FFM, fat-free mass; WC, waist circumference; FPG, fasting plasma glucose; FPI, fasting plasma insulin; RCT, randomized controlled trials.

**Table 2 ijerph-19-04638-t002:** Full-search strategy.

PICOS	Search Terms
Participant	obesity OR adult obesity OR adult with obesity OR obesity in adult OR obese OR overweight
Intervention	(High-Intensity Interval Training OR High Intensity Interval Training OR High-Intensity Interval Trainings OR Interval Training OR Interval Trainings OR High-Intensity Intermittent Exercise OR High-Intensity Intermittent Exercises OR Sprint Interval Training OR Sprint Interval Trainings OR high intensity sprint OR aerobic interval training OR aerobic interval trainings OR HIIT OR HIIE) AND (Fasting OR fasting OR Intermittent Fasting OR Intermittent Fastings OR Hunger Strike OR Hunger Strikes OR Time Restricted Feeding OR Time Restricted Feedings)
Study	randomized controlled trial OR randomized OR placebo
Outcome	body weight OR body mass OR BMI OR body mass index OR body fat percent OR percent fat mass OR body fat mass OR fat-free mass OR lean body mass OR VO_2peak_ OR fasting plasma glucose OR fasting plasma insulin

**Table 3 ijerph-19-04638-t003:** The characteristics of included RCTs.

Study	Country	Sample Size	Age Years	Types of Sport	HIIT Intervention	Fasting Intervention	Frequency	Duration	Control	Outcomes
Francois 2018	USA	23	46 ± 12	Bicycle	Exercise: 90%HR_max_ for 3 min; Intermittent: 50%HR_max_ for 3 min; 6–10 times	LCD (1000–1200 kcal/day)	HIIT: 7 times/week Fasting: 7 times/week	13 days	Only maintain LCD	BM; BMI; VO_2peak_; FM; PFM; FFM; FPG; FPI
Gilbertson 2019	USA	24	48.3 ± 13	Bicycle	Exercise: 90%HR_max_ for 3 min; Intermittent: 50%HR_max_ for 3 min; 6–10 times	LCD (1000–1200 kcal/day)	HIIT: 7 times/week Fasting: 7 times/week	2 weeks	Only maintain LCD	BM; BMI; WC; VO_2peak_; FM; PFM; FFM; FPG; FPI
Gillen 2013	Canada	16	27 ± 8	Bicycle	Exercise: 90%HR_max_ for 60 s; Intermittent: rest or pedal slowly at a resistance of 50 W for 60 s; 10 times	Keeping overnight fasting before exercise	HIIT: 3 times/week Fasting:3 times/week	6 weeks	Meal at 60min before HIIT	BM; FM; PFM; FPG; FPI
Gyorkos	USA	12	18–60	Bicycle	Exercise: 90%HR_max_ for 60 s; Intermittent: active recovery for 60 s; 10 times	A carbohydrate-restricted Paleolithic-based diet	HIIT: 3 times/week Fasting: 7 times/week	4 weeks	Only maintain a carbohydrate-restricted Paleolithic-based diet	BM; BMI; WC; PFM; VO_2peak_; FPG;FPI
Heiston 2020	USA	25	47.1 ± 12	Bicycle	Exercise: 90%HR_max_ for 3 min; Intermittent: 50%HR_max_ for 3 min; 6–10 times	LCD (1000–1200 kcal/day)	HIIT: 7 times/week Fasting: 7 times/week	13 days	Only maintain LCD	BM; BMI; WC; VO_2peak_; PFM; FFM; FPG; FPI
Malin 2020	USA	26	47.3 ± 12.2	Bicycle	Exercise: 90%HR_max_ for 3 min; Intermittent: 50%HR_max_ for 3 min; 6–10 times	LCD (1000–1200 kcal/day)	HIIT:7 times/week Fasting: 7 times/week	13 days	Only maintain LCD	BM; BMI; WC; PFM; FM; FFM; VO_2peak_
Pedersen 2019	Denmark	55	45–75	Bicycle	Exercise: 85–90%VO_2peak_ for 1–4 min; Intermittent: 65–70%VO_2peak_ for 1–3 min 4 times	LCD (800–1000 kcal/day)	HIIT: 3 times/week Fasting: 7 times/week	1 years	Only maintain HIIT	BM; BMI; FM; FFM; WC; VO_2peak_;
Sartor 2010	UK	19	39 ± 12	Bicycle	Exercise: 90%VO_2peak_ for 4 min; Intermittent: rest for 2–3 min; 10 times	A moderately low-carbohydrate and high-unsaturated fat diet	HIIT: 3 times/week Fasting: 7 time/week	2 weeks	Only maintain fasting intervention	BM; FM; FFM; FPG; FPI;
Sun 2019	China	30	21.2 ± 3.3	Bicycle	Exercise: Sprint phases for 6 s; Intermittent: passive recovery for 9 s; 10 times	A low-carbohydrate diet	HIIT: 5 times/week Fasting: 7 times/week	4 weeks	Maintain normal dietary habits and normal exercise habits	BM; BMI; VO_2peak_; WC; FPG;

BM, body mass; BMI, body mass index; FM, fat mass; PFM, percent fat mass; FFM, fat-free mass; WC, waist circumference; FPG, fasting plasma glucose; FPI, fasting plasma insulin; LCD, low-calorie diets.

**Table 4 ijerph-19-04638-t004:** Subgroup analysis for HIIT + fasting in overweight and obese adults.

Outcomes	Intervention	N	ES (95% CI)	Heterogeneity		*p*
				I^2^ (%)	*p*	
Body mass	Long term	4	MD: −4.28	0	0.45	0.008
			(−7.44, −1.13)			
	Short term	4	MD: 2.91	0	1	0.39
			(−3.78, 9.06)			
	Long interval	6	MD: −2.93	7	0.37	0.16
			(−6.99, 1.13)			
	Short interval	3	MD: −0.81	0	0.91	0.77
			(−6.20, 4.58)			
	LCD	6	MD: −2.31	12	0.34	0.3
			(−6.69, 2.07)			
	Low-carbohydrate diet	3	MD: −1.02	0	0.96	0.71
			(−6.5, 4.45)			
BMI	Long term	3	MD: −1.56	0	0.39	0.01
			(−2.76, −0.36)			
	Short term	4	MD: 0.36	0	0.98	0.78
			(−2.22, 2.94)			
	Long interval	5	MD: −1.23	0	0.7	0.03
			(−2.55, 0.09)			
	Short interval	2	MD: −1.43	40	0.2	0.29
			(−4.08, 1.22)			
	LCD	5	MD: −1.23	0	0.7	0.03
			(−2.55, 0.09)			
	Low-carbohydrate diet	2	MD: −1.43	40	0.2	0.29
			(−4.08, 1.22)			
WC	Long term	3	MD: −5.14	0	0.7	0.002
			(−8.33, −1.96)			
	Short term	3	MD: −1.73	0	0.76	0.55
			(−7.43, 3.97)			
	Long interval	4	MD: −4.22	0	0.65	0.01
			(−7.54, −0.89)			
	Short interval	2	MD: −4.6	0	0.42	0.07
			(−9.66, 0.46)			
	LCD	4	MD: −4.22	0	0.65	0.01
			(−7.54, −0.89)			
	Low-carbohydrate diet	2	MD: −4.6	0	0.42	0.07
			(−9.66, 0.46)			
BFP	Long term	3	MD: −1.04	0	0.67	0.26
			(−2.84, 0.76)			
	Short term	3	MD: −1.05	0	0.58	0.31
			(−3.09, 0.98)			
	Long interval	4	MD: −1.08	0	0.78	0.14
			(−2.5, 0.35)			
	Short interval	2	MD: −0.78	0	0.37	0.72
			(−5.02, 3.46)			
	LCD	5	MD: −0.96	0	0.8	0.18
			(−2.35, 0.43)			
	Low-carbohydrate diet	1	MD: −2.4	NA	NA	0.4
			(−7.95, 3.15)			
FM	Long term	2	MD: −2.47	2	0.31	0.01
			(−4.45, −0.5)			
	Short term	3	MD: 0.65	0	0.86	0.8
			(−4.42, 5.71)			
	Long interval	5	MD: −2.33	0	0.69	0.007
			(−4.02, −0.63)			
	Short interval	1	MD: 2	NA	NA	0.66
			(−6.94, 10.94)			
	LCD	5	MD: −2.45	0	0.54	0.01
			(−4.17, −0.73)			
	Low-carbohydrate diet	1	MD: −2	NA	NA	0.65
			(−10.59, 6.59)			
FFM	Long term	2	MD: −2.28	0	0.58	0.28
			(−6.38, 1.82)			
	Short term	5	MD: 1.57	0	0.99	0.28
			(−1.27, 4.41)			
	Long interval	6	MD: 0.48	0	0.74	0.7
			(−2.01, 2.98)			
	Short interval	1	MD: −0.8	NA	NA	0.81
			(−7.41, 5.81)			
	LCD	6	MD: 0.16	0	0.77	0.9
			(−2.24, 2.57)			
	Low-carbohydrate diet	1	MD: 3	NA	NA	0.55
			(−6.82, 12.82)			
VO_2peak_	Long term	3	SMD: 0.67	58	0.09	0.04
			(0.03, 1.32)			
	Short term	4	SMD: 0.85	29	0.24	0.0006
			(0.37, 1.34)			
	Long interval	5	SMD: 0.77	13	0.33	0.0001
			(0.41, 1.13)			
	Short interval	2	SMD: 0.89	77	0.04	0.15
			(−0.32, 2.09)			
	LCD	5	SMD: 0.77	13	0.33	0.0001
			(0.41, 1.13)			
	Low-carbohydrate diet	2	SMD: 0.89	77	0.04	0.15
			(−0.32, 2.09)			
FPG	Long term	3	SMD: −0.61	0	0.42	0.01
			(−1.09, −0.13)			
	Short term	4	SMD: 0.22	0	0.84	0.22
			(−0.19, 0.63)			
	Long interval	4	SMD: 0.22	0	0.84	0.22
			(−0.19, 0.63)			
	Short interval	3	SMD: −0.61	0	0.42	0.01
			(−1.09, −0.13)			
	LCD	4	SMD: 0.23	0	0.84	0.29
			(−0.19, 0.65)			
	Low-carbohydrate diet	3	SMD: −0.58	4	0.35	0.02
			(−1.06, −0.09)			
FPI	Long term	2	SMD: −0.05	0	0.97	0.87
			(−0.67, 0.57)			
	Short term	4	SMD: 0.21	0	0.7	0.32
			(−0.2, 0.62)			
	Long interval	4	SMD: 0.21	0	0.7	0.32
			(−0.2, 0.62)			
	Short interval	2	SMD: −0.05	0	0.97	0.87
			(−0.67, 0.57)			
	LCD	4	SMD: 0.17	0	0.64	0.43
			(−0.25, 0.59)			
	Low-carbohydrate diet	2	SMD: 0.05	0	0.74	0.87
			(−0.55, 0.65)			

**Table 5 ijerph-19-04638-t005:** Summary of findings.

Patient or population: patients with overweight and obese adults Settings: HIIT + fasting compared to other intervention in the treatment of overweight and obese adults Intervention: HIIT + fasting Comparison: HIIT alone or fasting alone or normal intervention				
Outcomes	Illustrative comparative risks * (95% CI)	Relative effect (95% CI)	No of Participants (studies)	Quality of the evidence (GRADE)
	Corresponding risk			
	HIIT + fasting			
Body mass	The mean body mass in the intervention groups was 2.97 lower (5.83 to 0.12 lower)		242 (9 studies)	⊕⊝⊝⊝ very low ^1,2,3^
BMI	The mean BMI in the intervention groups was 1.22 lower (2.31 to 0.13 lower)		207 (7 studies)	⊕⊝⊝⊝ very low ^1,2,3^
WC	The mean wc in the intervention groups was 4.33 lower (7.11 to 1.55 lower)		184 (6 studies)	⊕⊝⊝⊝ very low ^1,2,3^
PFM	The mean percentage fat mass in the intervention groups was 1.05 lower (2.4 lower to 0.3 higher)		169 (6 studies)	⊕⊝⊝⊝ very low ^1,2,3,4^
FM	The mean fat mass in the intervention groups was 2.18 lower (3.84 to 0.51 lower)		163 (6 studies)	⊕⊝⊝⊝ very low ^1,2,3^
FFM	The mean free fat mass in the intervention groups was 0.32 higher (2.01 lower to 2.66 higher)		193 (7 studies)	⊕⊝⊝⊝ very low ^1,2,3,4^
VO_2peak_	The mean vo2 in the intervention groups was 0.78 SD higher (0.42 to 1.14 higher)		207 (7 studies)	⊕⊝⊝⊝ very low ^1,2,3^
FPG	The mean fpg in the intervention groups was 0.12 SD lower (0.51 lower to 0.27 higher)		162 (7 studies)	⊕⊝⊝⊝ very low ^1,2,3,4^
FPI	The mean fpi in the intervention groups was 0.13 SD higher (0.21 lower to 0.47 higher)		132 (6 studies)	⊕⊝⊝⊝ very low ^1,2,3,4^

* The basis for the assumed risk (e.g., the median control group risk across studies) is provided in footnotes. The corresponding risk (and its 95% confidence interval) is based on the assumed risk in the comparison group and the relative effect of the intervention (and its 95% CI). CI: Confidence interval; SD: Standard deviations; BM, body mass; BMI, body mass index; FM, fat mass; PFM, percent fat mass; FFM, fat-free mass; WC, waist circumference; FPG, fasting plasma glucose; FPI, fasting plasma insulin. GRADE Working Group grades of evidence. High quality: Further research is very unlikely to change our confidence in the estimate of effect. Moderate quality: Further research is likely to have an important impact on our confidence in the estimate of effect and may change the estimate. Low quality: Further research is very likely to have an important impact on our confidence in the estimate of effect and is likely to change the estimate. Very low quality: We are very uncertain about the estimate. ^1^ The allocation concealment of some included studies was unclear. There was no blinding for participants and personnel in some included studies. The outcome assessment of some included studies was unclear. Final decision: we lowered the study limit by one level. ^2^ Although *p* > 0.05 and I^2^ = 0%, the fasting plans of studies were different. Final decision: we lowered the study limit by one level. ^3^ Number of participants < 400. Final decision: we lowered the study limit by one level. ^4^ The confidence interval reached the null effect. Final decision: we lowered the study limit by one level.

## Data Availability

All the included studies are in Table 3.

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
