# Peer review of "Effect of High-Intensity Interval Training Combined with Fasting in the Treatment of Overweight and Obese Adults: A Systematic Review and Meta-Analysis"

_ijerph, 2022, doi:10.3390/ijerph19084638_

Round 1

Reviewer 1 Report

The  paper presents the  meta-analysis  of  the impact of  high intensity interval training  combined with  fasting  on the overweight and obesity.

The authors may consider following comments:

  1. The described process of systematic review and meta-analysis seems to be correct. Conclusions drawn from the collected material (even though it appears to be quite sparse - only 9 articles meet the entry criteria) are consistent with the knowledge in the studied subject.
  2. In the abstract section: what was the idea behind presenting some data as MD and some as SMD?
  3. The part concerning citation needs to be harmonized. It is suggested that all footnotes have Arabic numerals.
  4. Table 1 presents the  track of  looking for   appropriate  Authors may consider whether it is needed to be presented in that form. It is generally difficult to follow and takes a lot of space.
  5. Some sentences in the text need to be analyzed and reformatted. They are long and require breakdown / shortening / simplification.
  6. In addition, it is worth analyzing the text in terms of repetitions, e.g. combined fasting in obese adults, and there is no consensus as to whether HIIT combined with fasting (HF) is superior to HIIT alone or fasting alone. Therefore we aim to quantify the impact of HIIT combined fasting.
  7. Please remove capital letters from inside sentences.
  8. Please complete the description of all abbreviations used in the work i.e. MICT
  9. If HITT + fasting means HF it is advised to change it in the text to make it 
  10. Verse 6: “fast author’s last name” do the authors have in mind first author’s last name?
  11. Statistical analysis section: as it concerns the activities performed, it is reasonable to use the past tense instead of the future tense.
  12. Table 2 horizontally oriented would be much clearer.
  13. Figure 4 5 6 - why have they been divided into three separate figures since they are homogenic with respect to the presented data? perhaps it would be worthwhile to incorporate the material into a single object.
  14. please divide table 3 into subsections - so that the reader quickly finds out where the analysis of one variable ends and the description of the next one begins.
  15. Line 126: what “ diety” stands for?

Summary: The article in its current form requires, above all, a thorough linguistic improvement. In terms of methodology and content, the work seems to be correct.

Author Response

Thank you for your comments. The response is shown in the attachment.

Reviewer 2 Report

ABSTRACT

On the objectives, add more information about participants, thus providing some information about adults (all sedentary?). Moreover, it is not clear the main outcomes analyzed. The effects of HIIT and fasting on which main outcomes? Specify based on PICOS.

In the methods of the abstract specify the inclusion and exclusion criteria for the review; specify the methods used to assess the risk of bias in the included studies, and Specify the methods used to present and synthesize results.

The conclusions provide a brief summary of the limitations of the evidence included in the review (e.g. study risk of bias, inconsistency, and imprecision).

INTRODUCTION

Line 1 of paragraph 1: define obesity based on normative values for BMI and/or fat mass.

Paragraph 2: the two sentences were cut by a “.”. Moreover, support the statements with clinical evidence and references.

Paragraph 2: specify which type of exercise. Any exercise? Or intense exercise? Consider reading about physiological mechanisms to explain that effect.

Here are some documents to read aiming to improve the section:

Maillard, F., Pereira, B., & Boisseau, N. (2018). Effect of high-intensity interval training on total, abdominal and visceral fat mass: a meta-analysis. Sports Medicine48(2), 269-288.

Viana, R. B., Naves, J. P. A., Coswig, V. S., De Lira, C. A. B., Steele, J., Fisher, J. P., & Gentil, P. (2019). Is interval training the magic bullet for fat loss? A systematic review and meta-analysis comparing moderate-intensity continuous training with high-intensity interval training (HIIT). British journal of sports medicine.

Wewege, M., Van Den Berg, R., Ward, R. E., & Keech, A. (2017). The effects of high‐intensity interval training vs. moderate‐intensity continuous training on body composition in overweight and obese adults: a systematic review and meta‐analysis. Obesity Reviews18(6), 635-646.

Paragraph 2: my suggestion is to split the paragraph and organize it into two: one related to exercise and the other with nutritional strategies.

METHODS

Start the methods with PICOS (Population, intervention, comparator, outcome, study design) details.

Inclusion and exclusion criteria section: add a table with three columns: One for PICOS, second for inclusion, and third for exclusion. Add lines for each section of PICOS. After that, fill out the cells with the description of each criterion based on each topic of PICOS. Consider providing rationales for any notable restrictions to study eligibility.

Before the risk of bias add the section:

DATA ITEMS (outcomes): • List and define the outcome domains and time frame of measurement for which data were sought. • Specify whether all results that were compatible with each outcome domain in each study were sought, and if not, what process was used to select results within eligible domains. • If any changes were made to the inclusion or definition of the outcome domains, or to the importance given to them in the review, specify the changes, along with a rationale. • If any changes were made to the processes used to select results within eligible outcome domains, specify the changes, along with a rationale. • Consider specifying which outcome domains were considered the most important for interpreting the review’s conclusions and provide rationale for the labelling (e.g. “a recent core outcome set identified the outcomes labelled ‘critical’ as being the most important to patients”). And • List and define all other variables for which data were sought (e.g. participant and intervention characteristics, funding sources). • Describe any assumptions made about any missing or unclear information from the studies. • If a tool was used to inform which data items to collect, cite the tool used.

Risk of bias: provide details about the name and instrument used. Have you used ROB-2 probably. Present the main topics covered and the validity and reliability of the instrument.  

After statistical analysis add:

CERTAINTY ASSESSMENT: • Specify the tool or system (and version) used to assess certainty (or confidence) in the body of evidence. • Report the factors considered (e.g. precision of the effect estimate, consistency of findings across studies) and the criteria used to assess each factor when assessing certainty in the body of evidence. • Describe the decision rules used to arrive at an overall judgement of the level of certainty, together with the intended interpretation (or definition) of each level of certainty. • If applicable, report any review-specific considerations for assessing certainty, such as thresholds used to assess imprecision and ranges of magnitude of effect that might be considered trivial, moderate or large, and the rationale for these thresholds and ranges (item #12). If any adaptations to an existing tool or system to assess certainty were made, specify the adaptations. • Report how many reviewers assessed certainty in the body of evidence for an outcome, whether multiple reviewers worked independently, and any processes used to resolve disagreements between assessors. • Report any processes used to obtain or confirm relevant information from investigators. • If an automation tool was used to support the assessment of certainty, report how the automation tool was used, how the tool was trained, and details on the tool’s performance and internal validation. • Describe methods for reporting the results of assessments of certainty, such as the use of Summary of Findings tables. • If standard phrases that incorporate the certainty of evidence were used (e.g. “hip protectors probably reduce the risk of hip fracture slightly”), report the intended interpretation of each phrase and the reference for the source guidance

RESULTS

Line 43: add the reasons for exclusion of full-text articles and add the references associated with those exclusions.

Line 68: describe the main bias identified across the articles.

PLEASE ADD CERTAINTY OF EVIDENCE: • Report the overall level of certainty (or confidence) in the body of evidence for each important outcome. • Provide an explanation of reasons for rating down (or rating up) the certainty of evidence (e.g. in footnotes to an evidence summary table). • Communicate certainty in the evidence wherever results are reported (i.e. abstract, evidence summary tables, results, conclusions), using a format appropriate for the section of the review. • Consider including evidence summary tables, such as GRADE Summary of Findings tables.

DISCUSSION

At the bottom of the discussion, add a section of future research needed, the main bias of the included articles, and possible bias effects on the main evidence. Moreover, discuss implications of the results for practice and policy.

Author Response

(The authors gave the same response as above.)

Reviewer 3 Report

Overall

  • Please number the lines
  • Please avoid the unnecessary use of abbreviations. Please try to remove most of them from both abstract and main text.
  • English language needs careful revision. Grammar is overall fine, but syntax and fluency need to be revisited.
  • Please check the references format.
  • Please use body mass instead of body weight.

Abstract

  • What did the result compare with? NO training? Training alone? Fasting alone? Please specify.

Introduction

The introduction should be improved in term of providing a rationale for the present meta-analysis. The first paragraph could be abbreviated, and the effects of HIIT should be introduced more in depth. Same for fasting. Additionally, at the end of the introduction I am not sure what the meta-analysis wants to compare. Please re-organize the introduction to clearly introduce the research question.

Methods

  • Only articles published in English language should be considered. It seems this does not affect the final result, just change the flow-chart steps.
  • Please present table 1 in a readable way.

Results

As for the abstract, I don’t understand the pairwise comparison between the HIIT+ fasting vs control group, mostly because it’s not clear what the control group did (i.e., HIIT alone, fasting alone, nothing). Separate analysis should be performed to let me understand the actual effectiveness of the combined use of HIIT+fasting vs the other methods.

Discussion

The discussions suffers from the previous concerns.

Author Response

(The authors gave the same response as above.)

Round 2

Reviewer 1 Report

Line 10-> inside a sentence capital letter

 Line 20->  The risk of bias was assessed by the Cochrane risk of bias tool.

Line 35 Conclusions: Despite the limitation on the number of included trials and the GRADE of all outcomes was very low. This sentence is not clear.

 Lines 58-61 Repetitions:

Exercise is currently recognized as one of the best measures to treat obesity. 58 because it can improve body composition and enhance exercise capacity. There are many[11]. Many researches that documented that moderate exercise has a positive effect on losing weight, reducing central adiposity, and preventing obesity[12][13][14]. A recent systematic review reveals that exercise

Thesaurus  examples: training,  physical activity/ inactivity, aerobics

 Line 70 -> inside a sentence capital letter

Line 115 -> inside a sentence capital letter

Line 195-> fast author’s last name,

Line 207 -> and the risk of bias assessment of the included RCTs was assessed using a “risk of  bias” approach which is recommended by the Cochrane guidelines risk of bias tool.

 Please guys,  try to not “ assess the assessment”.  Construct as simple sentences  as possible, avoid repetitions, then it will be readable.

Table 4 – I would provide   horizontal lines between subgroups.

 Table 5  - columns  assumed risk  relative effect and comments are empty. Is it how it should be?

Author Response

Thank you very much for your careful review. Please see the attachment.

Reviewer 2 Report

The article was meaningfully improved and, based on that, I would like to recommend the acceptance

Author Response

Thank you very much for your careful review and approval.

Reviewer 3 Report

No further comment

Author Response

(The authors gave the same response as above.)
